# Pulsed Electric Field Pretreatments Affect the Metabolite Profile and Antioxidant Activities of Freeze− and Air−Dried New Zealand Apricots

**DOI:** 10.3390/foods13111764

**Published:** 2024-06-04

**Authors:** Ye Liu, Indrawati Oey, Sze Ying Leong, Rothman Kam, Kevin Kantono, Nazimah Hamid

**Affiliations:** 1Centre for Future Foods, Auckland University of Technology, Private Bag 92006, Auckland 1142, New Zealand; ye.liu@aut.ac.nz (Y.L.); rothman.kam@aut.ac.nz (R.K.); kkantono@aut.ac.nz (K.K.); 2Department of Food Science, University of Otago, PO Box 56, Dunedin 9054, New Zealand; indrawati.oey@otago.ac.nz (I.O.); sze.leong@otago.ac.nz (S.Y.L.)

**Keywords:** PEF, metabolites, freeze−drying, air−drying, apricot

## Abstract

Pulsed electric field (PEF) pretreatment has been shown to improve the quality of dried fruits in terms of antioxidant activity and bioactive compounds. In this study, apricots were pretreated with PEF at different field strengths (0.7 kV/cm; 1.2 kV/cm and 1.8 kv/cm) at a frequency of 50 Hz, and electric pulses coming in every 20 µs for 30 s, prior to freeze−drying and air−drying treatments. PEF treatments were carried out at different field strengths. The impact of different pretreatments on the quality of dried apricot was determined in terms of physical properties, antioxidant activity, total phenolic content, and metabolite profile. PEF pretreatments significantly (*p* < 0.05) increased firmness of all the air−dried samples the most by 4–7−fold and most freeze−dried apricot samples (44.2% to 98.64%) compared to the control group. However, PEF treatment at 1.2 kV/cm did not have any effect on hardness of the freeze−dried sample. The moisture content and water activity of freeze−dried samples were found to be significantly lower than those of air−dried samples. Scanning electron microscopy results revealed that air drying caused the loss of fruit structure due to significant moisture loss, while freeze drying preserved the honeycomb structure of the apricot flesh, with increased pore sizes observed at higher PEF intensities. PEF pretreatment also significantly increased the antioxidant activity and total phenol content of both air−dried and freeze−dried apricots. PEF treatment also significantly (*p* < 0.05) increased amino acid and fatty acid content of air−dried samples but significantly (*p* < 0.05) decreased sugar content. Almost all amino acids (except tyrosine, alanine, and threonine) significantly increased with increasing PEF intensity. The results of this study suggest that PEF pretreatment can influence the quality of air−dried and freeze−dried apricots in terms antioxidant activity and metabolites such as amino acids, fatty acids, sugar, organic acids, and phenolic compounds. The most effective treatment for preserving the quality of dried apricots is freeze drying combined with high−intensity (1.8 kv/cm) PEF treatment.

## 1. Introduction

Apricots (*Prunus armeniaca* L.) can be grown in the temperate zones of the world [1,2] They were originally grown on wild Chinese hillsides and later cultivated in Armenia [3] In 2020, the worldwide apricot cultivation area was 562,475 hectares, and the total apricot yield amounted to 3.72 million metric tons [4]. The leading producer of apricots is Turkey, with other major contributors being Uzbekistan, Iran, Algeria, Italy, Afghanistan, Spain, Greece, Pakistan, and Morocco. [5]. Apricots are rich in vitamin A precursors [6]. However, apricots are seasonal fruits that are highly perishable and experience huge losses throughout the production chain [7]. Drying is a typical preservation technique for extending food shelf life and ensuring food security. Apricots are traditionally pretreated with sulphur dioxide before sun drying to preserve their natural reddish yellow colour [8]. However, hot air drying involves long drying times and often causes undesirable quality issues. Freeze drying on the other hand can maintain the fruit quality, but its high energy consumption requirement significantly increases cost [7].

Pulsed electric field (PEF) is a non−thermal technology that has been used as a pretreatment prior to drying of fruits. It can increase cell permeabilisation, which leads to reduced drying time, and can release bioactive components in plant tissues [9]. Cell membrane permeability can lead to the release of intracellular compounds. Depending on the fruit structure, this may either increase the availability of antioxidants or cause their degradation. These bioactive components can increase the nutritional value of dried fruit. In addition, PEF can reduce the degradation of heat−sensitive compounds. 

Previous studies have mainly focused on the effect of PEF on drying of fruits in terms of antioxidant activity, drying efficiency, and changes in physical properties. PEF pretreatment at low energy levels has been reported to improve the quality of freeze−dried red bell peppers and strawberries in terms of rehydration capacity, shrinkage reduction and texture [10]. A combination of osmotic dehydration (55 °C, 60 min), PEF (2.8 kV/cm, 750 p) and air−drying (60 °C) treatment compared to conventional air−dried goji berry decreased total processing time by 180 min (33%), improved colour retention, as well as increased antioxidant capacity and total phenolic content [11]. PEF treatments at a specific setting (1.25 kV/cm, 100 Hz, 20 µs) led to an increase in the drying rate, β−carotene content and antioxidant activity in air−dried apricots [7]. Previous study further observed that the moisture content in freeze−dried apples subjected to PEF treatment (1.07 kV/cm, 2 Hz) was reduced by about 82% compared to untreated apples [12].

PEF treatment prior to drying can influence antioxidant activity of fruits. Air−drying (70 °C) apple tissue subjected to PEF pretreatments (3.5 and 6 kJ/kg) led to a significant decrease (*p* < 0.05) in antioxidant activity using DPPH and iron−ion−reducing power assays [13]. Similarly, PEF pretreatment (1.0 and 3.0 kJ/kg) prior to convective and microwave–convective drying of red bell pepper resulted in a significant (*p* < 0.05) decrease in antioxidant activity, as determined by DPPH and ATBS assays [14]. However, PEF pretreatments applied to mango peels (at 1.5 kV cm^−1^, 3.0 kV cm^−1^ and 4.5 kV cm^−1^) at varying air−drying temperatures (50, 60 and 70 °C) resulted in a significant (*p* < 0.05) increase in antioxidant activity with increasing temperature. Moreover, the combination of 70 °C and 4.5 kV cm^−1^ yielded the highest antioxidant activity, as evaluated using DPPH, ABTS and FRAP assays [15].

PEF treatment can strongly influence the tissue firmness of fruit and vegetables through electroporation [16]. This phenomenon is attributed to changes in cell membrane permeability arising from the effect of PEF on cell structure [17]. Hot air−dried kiwifruit pretreated with PEF (1.92 kJ/kg) had significantly (*p* < 0.05) decreased firmness compared to the control group [17] Similar findings were reported for freeze−dried strawberries and bell peppers, where PEF pretreatments at 1.5, 3 and 6 kJ/kg significantly (*p* < 0.05) decreased firmness [10]. Moreover, the specific energy input and storage time (30 days) did not have a significant effect on hardness compared to the control group. On the other hand, PEF−pretreated hot air−dried pumpkin showed a significant (*p* < 0.05) increase in hardness compared to the control group. Similarly, PEF treatment of carrots and parsnips at 65.2 ± 0.7 kJ/kg and 65.8 ± 1.6 kJ/kg, respectively, significantly (*p* < 0.05) increased in hardness when dried at 70 °C [18].

PEF can be used to increase the extraction of polyphenols from fruits. Previous studies have shown that PEF treatment can enhance the migration of polyphenol compounds from cells to the surrounding matrix and prevent polyphenol degradation by inactivation of polyphenoloxidase [19,20,21,22].The study on the effect of PEF treatment on major polyphenol compounds in *Rosa canina* fruits found that PEF treatments at 1.7 kv/cm 0.389 kJ/kg and 2.0 kv/cm 0.457 kJ/kg significantly (*p* < 0.05) increased quercetin 3−O−glucoside, quercetin 3−O−rutinoside, eriodictyol 7−O−rutinoside and catechin content in these fruits compared to control samples [20]. Similarly, PEF treatments at 1.7 kv/cm 0.389 kJ/kg and 2.0 kv/cm 0.457 kJ/kg significantly (*p* < 0.05) increased quercetin 3−rutinoside, quercetin 3−glucoside, kaempferol 3−glucoside, quercetin 3−glucuronide and gallic acid in *Vitis vinifera* fruit compared to control samples [21]. PEF treatment (3 kV/cm; 5, 10 and 15 kJ/kg) of cranberry bush purée has also been found to significantly increase (*p* < 0.05) chlorogenic acid compared to control [19]. Although PEF treatment can significantly increase the levels of some polyphenol compounds in fruit at high specific energy levels, the enhancement effect may plateau or decrease at very high energy input levels. Previse study found that PEF treatment of orange peel at electric field strengths of 1 to 7 kV/cm (0.06 to 3.77 kJ/kg) significantly increased the extraction of naringin and hesperidin compared to the untreated sample (*p* < 0.05) [22]. The extraction of both flavonoids increased with the intensity of the electric field strength until 5 kV/cm, after which there was no significant difference (*p* > 0.05) between the 5 and 7 kV/cm treatments [22].

Given that both amino acids and sugars are involved in metabolic processes and can be derived from or transformed into other metabolic products, they are classified as metabolites in the broader context of biochemistry and metabolism. In terms of changes in metabolites, previous studies have shown that PEF can increase amino acids and sugar content. Previous study reported that the total free amino acids were significantly (*p* < 0.05) higher in grape juice treated by PEF compared to control [23]. However, PEF treatments showed no significant effects on fatty acids in this study The concentrations of essential amino acids (arginine, histidine, leucine, isoleucine, lysine, phenylalanine, and threonine) were significantly (*p* < 0.05) higher in samples treated at a field strength of 2.92 kV/cm compared to control. PEF processing can significantly increase the total sugar content of juices from black chokeberry [24]. The higher sugar content may be attributed to the increase in dry matter after PEF processing. However, the effects of PEF pretreatment on the metabolite profiles of fruits prior to drying have not been fully investigated.

The effects of PEF treatment on the physicochemical properties of different dried fruits are complex and diverse. Further research is needed to understand the parameters of PEF pretreatments for improving the quality and nutritional value of dried fruits. Hence, this study investigated the changes in microstructural, physical, antioxidant activity and metabolite profiles of PEF−pretreated apricots subjected to freeze and air drying. The results of this study provide valuable insights into the potential of PEF pretreatment to improve the physical and chemical characteristics of dried fruits.

## 2. Materials and Methods

### 2.1. Apricot Samples

Fresh apricots were purchased from Bidfood, Dunedin, New Zealand in February 2020. These apricots belonged to the Sundrop cultivar that was obtained from Roxburgh, South Island, New Zealand. Apricots with similar size and weight were selected for the experiment. In total, twenty−four apricot fruits were used in this study.

### 2.2. PEF Treatment and Drying

PEF processing was carried out using the ELCRACK HVP apparatus designed by the German Institute of Food Technologies, Quakenbruck, Germany. The apparatus was a batch mode reactor. Hence, PEF treatment for the apricot samples was carried out in two batches with eight replicates each. The PEF chamber dimensions were 100 mm in length, 80 mm in width and 50 mm in height. Only two fruits were treated per batch at a time. Four different field strength levels of PEF pretreatments were carried out in this study: (i) no PEF, (ii) low PEF (0.7 kV/cm), (iii) medium PEF (1.2 kV/cm) and (iv) high PEF (1.8 kV/cm), with a frequency of 50 Hz and electric pulses of 20 µs for 30 s. These PEF parameters were chosen based on the desired effect of different PEF intensities. The induction of the stress response was achieved with intensities of 0.5–1.5 kV/cm and energy inputs of 0.5–5 kJ/kg. The improvement of mass transfer was targeted with intensities of 0.7–3 kV/cm and energy inputs of 1–20 kJ/kg [25]. The control apricot samples with no PEF treatment were submerged in a water bath for 30 s with no pulse electric fields passing through these samples.

The PEF−treated apricot samples were cut in half and pitted (removal of the seed) after PEF processing. One half of the apricot samples were subjected to air drying and the other half to freeze drying. Directly after PEF treatment, the apricot samples were air dried at 70 °C for 12 h. For freeze drying, the apricot samples were frozen directly after PEF treatment using a blast chiller (Irinox MultiFresh^®^ blast chiller MF 45.1, Madonna di Loreto, 6/B, Corbanese, Italy) for 12 h at −20 °C. Then, the samples were moved to a VirTis 35L general purpose freeze dryer (VirTis SP scientific sentry 2.0; SP Industries, 935 Mearns Road, Warminster, PA, USA). The entire procedure lasted for 72 h, with the vacuum set at 0.05 mbar. During the sublimation phase, the temperature was maintained at approximately 0 °C, while in the secondary drying phase, the temperature was increased to 30 °C. Sample treatments used in this study are summarised in Table 1.

### 2.3. Physical Analysis

#### 2.3.1. Texture Evaluation

The texture evaluation of dried apricot involved using a TA.XTplus texture analyzer equipped with a Film Support Rig (HDP/FSR) on a heavy−duty platform (HDP/90), with all tests conducted at a constant temperature of 22 °C. Three samples of apricot were analyzed for each treatment, resulting in a total of 24 measurements. The texture parameters measured included hardness, adhesiveness, resilience, cohesion, springiness, gumminess, and chewiness. Penetrometer tests were carried out with a cylindrical probe of 2 mm in diameter, with penetration of samples to a depth of 10 mm at a speed of 0.5 mm/s. Additionally, compression tests used a 21 mm compression plate (P/50). The texture analyzer was configured in compression mode, with a pretest speed of 2.0 mm/s, a test speed of 1.0 mm/s and a posttest speed of 10.0 mm/s. The target mode of the texture analyzer was set to measure at a depth of 5 mm.

#### 2.3.2. Scanning Electron Microscopy Imaging Analysis

The micromorphology of dried apricots was analyzed using a Hitachi SU−70 scanning electron microscope (SEM). The detector was positioned at a working distance of 15.5 mm from the samples, with an accelerating voltage of 5 kV applied to each sample. A platinum coating was applied to the samples before scanning using the Hitachi E−1045 Ion Sputter. For visualization, the micrographs were represented with a 1 mm scale for the skin and a 100 μm scale for the pulp.

#### 2.3.3. Moisture Content and Water Activity

Moisture content was determined by subtracting the weight of the dried apricot sample from the weight of the fresh apricot sample. Water activity was measured using an AquaLab water activity meter (Series 3TE, Decagon Devices Inc., Pullman, WA, USA). Each measurement was performed in triplicate.

### 2.4. Chemical Analysis

#### 2.4.1. Sample Extraction Prior to Antioxidant Analysis

An apricot sample (0.1 g) was weighed in a centrifuge tube and extracted with 4 mL 50% methanol. The sample was homogenized for 1 min using a vortex and then centrifuged at 1500 rpm for 15 min at 4 °C. The supernatant liquid was extracted and transferred into a 10 mL volumetric flask. The same step was repeated using 70% acetone. After that, deionized water was added to the volumetric flask containing methanol and acetone extracts and water added to the 10 mL mark and mixed completely. The samples were further diluted by a factor of 10. The extracts were stored in a freezer at −20 °C to minimize sample oxidation.

#### 2.4.2. Cupric−Reducing Antioxidant Capacity (CUPRAC)

The cupric−reducing antioxidant capacity (CUPRAC) assay measures the antioxidant capacity of sample extracts, following the method described by Apak et al. [26]. In this assay, ascorbic acid (0.1 g) was dissolved and diluted to 100 mL to create a 1 g/L stock solution. Using this stock solution, standard solutions were prepared with final concentrations of 80, 40, 20, 10, 5, 2.5 and 0 mg/L. The absorbance at 450 nm was measured to generate a standard curve.

To test antioxidant capacity, 1 mL of the sample extract or distilled water was combined with 1 mL of 0.01 M CuCl_2_, 1 mL of 1 M NH_4_AC (pH 7), 1 mL of 0.075 M neocuproine and 0.1 mL of deionized water, resulting in a 4.1 mL total volume. The mixture was incubated for 5 min at room temperature, and then absorbance at 450 nm was measured using an Ultrospec 2100 Pro (Amersham Pharmacia Biotech, 800 Centennial Ave, Piscataway, NJ, USA) spectrophotometer. The antioxidant capacity was expressed as mg ascorbic acid equivalent per gram of sample. All measurements were performed in triplicate.

#### 2.4.3. Ferric−Reducing Antioxidant Power (FRAP)

The ferric−reducing antioxidant power (FRAP) assay was applied as described by Morais et al., [27]. Ascorbic acid (0.1 g) was dissolved in water and diluted to 100 mL to prepare a 1 g/L stock solution. From this stock solution, standard solutions with concentrations of 80, 40, 20, 10, 5, 2.5 and 0 mg/L were prepared.

The FRAP reagent was prepared by mixing 10 mL of 300 mM acetate buffer, 1 mL of 10 mM TPTZ (dissolved in 10 mL of 40 mM HCl) and 1 mL of 20 mM FeCl_3_, incubating at 36 °C in a water bath until an orange−yellow color appeared. Afterwards, 2 mL of the FRAP reagent was added to 0.1 mL of the sample extract or distilled water, with an additional 0.9 mL of distilled water. The mixture was allowed to rest for 4 min, and the absorbance at 593 nm was measured using an Ultrospec 2100 Pro spectrophotometer. The FRAP results were expressed as mg ascorbic acid equivalent per gram of sample. Like the CUPRAC assay, the FRAP determinations were also performed in triplicate.

#### 2.4.4. Total Phenolic Content (TPC)

The total phenolic content was determined using the Folin–Ciocalteau method by Singleton et al. [28]. The air−dried and freeze−dried samples were analyzed in triplicate. The absorbance of the samples was obtained by using the FLUOstar^®^ Omega spectrophotometer. A calibration curve was used to determine the antioxidant activity of the dried apricot samples and results were expressed in terms of ascorbic acid or Trolox equivalent in mg/g.

#### 2.4.5. Metabolite Analysis

For metabolite analyses, two derivatisation methods were used: trimethylsilylation (TMS) and methylchloroformate (MCF). TMS and MCF derivatives were prepared [29]. Gas chromatography–mass spectrometry (GC−MS) was utilized to identify, and semi−quantify organic acids, amino acids (except arginine) and fatty acids. The parameters used for GC−MS were based on the study by Smart et al. [30].

##### TMS Derivatisation

The dried sample was resuspended in 80 μL of methoxyamine hydrochloride solution in pyridine (2 g/100 mL). The sample was then incubated in a microwave oven for 2.8 min, with multimode irradiation set to 400 W at 30% power output. After that, 80 μL of N−methyl−N−(trimethylsilyl) trifluoroacetamide (MSTFA) was added to the sample, which was subsequently incubated for 3 min in the microwave oven.

##### MCF Derivatisation

The dried sample was resuspended in 200 μL of a 1 M sodium hydroxide solution, to which 34 μL of pyridine and 167 μL of methanol were added. The mixture was vigorously mixed for 30 s after adding 20 μL of MCF, followed by another 30 s vigorous mix after a second addition of 20 μL of MCF. Chloroform (400 μL) was added to separate the MCF derivatives from the reaction mixture, followed by vigorous mixing for 10 s. The solution was then treated with 400 μL of 50 mM sodium bicarbonate solution, followed by another vigorous mix for 10 s. The upper aqueous layer was discarded, leaving the chloroform phase for further GC−MS analysis.

### 2.5. Statistical Analysis

Results for antioxidant activity, total phenolic content, water activity, moisture content and texture were subjected to statistical analysis using the XLSAT MX software release 2010 (Addinsoft, New York, NY, USA). The data were analysed using one−way ANOVA to examine the main effects of PEF pretreatment on apricot samples. The data were later subjected to two−way ANOVA to examine the difference between freeze−dried and air−dried apricots pretreated with PEF prior to drying. When ANOVA was significant (*p* values < 0.05 and/or 0.1), means were separated by pairwise comparison using the Fisher’s least significant difference test.

Heat maps for showing the intensity of different metabolites in PEF−treated and non−PEF−treated dried apricots were generated by using MetaboAnalyst (version 4.0). Heatmaps were combined with hierarchical clustering, a technique that organizes items into a hierarchical structure based on the distance or similarity between them. The outcome of a hierarchical clustering analysis is presented in a heatmap through a dendrogram, which represents the hierarchical tree structure. Sparse partial least squares–discriminant analysis (sPLS−DA) was applied to summarize the metabolite differences between the control and the PEF−pretreated (at low, medium and high PEF intensities) apricot samples that were subjected to either air drying or freeze drying using MetaboAnalyst4.0. Results from metabolite profiling were analysed separately for the different drying methods using a one−way analysis of variance (ANOVA) using the XLSAT MX software release 2010 (Addinsoft, USA).

## 3. Results and Discussion

Biological variability of the samples was assessed using multivariate ANOVA (MANOVA) with Wilks’ test and Rao’s approximation where sample and replicate were included as main and interaction factors. No significant difference was observed in the repetition factor and its interaction.

### 3.1. Physical Analysis

#### Moisture Content and Water Activity

Table 2 summarizes the moisture content, water activity, texture, antioxidant ability and total phenolic content of PEF−pretreated apricots at either 0.7, 1.2 or 1.8 kV/cm at 50 hz every 20 µs for 30 s that were subsequently subjected to air drying or freeze drying. The results of our study indicate that the moisture content and water activity values of freeze−dried samples were significantly lower (*p* < 0.05) than those of air−dried samples. Specifically, the moisture content and water activity values of air−dried samples were found to be, on average, 28% and 30% higher than those of freeze−dried samples, respectively. Similarly, previous study reported that the moisture content of air−dried pumpkin (*Cucurbita moschata* Duch) flour was 13% higher compared to freeze−dried samples [31].

PEF pretreatment did not have a significant impact on the moisture content and water activity of the samples, regardless of the dehydration method used. There were no significant differences in terms of moisture content and water activity between the control samples and those treated with PEF (ranging from 0.7 kV/cm to 1.8 kV/cm) and subsequently freeze dried or air dried (see Table 2). For the samples subjected to air drying, the moisture content and water activity values ranged from 27.231% to 27.47% and 0.544 to 0.571, respectively. For the samples subjected to freeze drying, the moisture content and water activity values ranged from 19.42% to 19.853% and 0.384 to 0.399, respectively.Previous study reported that goji berries pretreated with PEF and osmotic dehydration at electric field strengths ranging from 0.9 kV/cm to 1.8 kV/cm and air dried had no significant effect on water loss and water activity. Sotelo et al. (2018) also found that PEF treatments using electric field strengths ranging from 0.3 kV/cm to 2.5 kV/cm had no significant effect on the moisture content of red cherries [11]. In addition, the Aw values of apricot samples dried using both freeze−drying and air−drying methods after PEF pretreatment were all below 0.6. This suggests that there will be no microbial proliferation with storage, as reported by Fontana [32].

### 3.2. Texture

Texture of dried apricot samples subjected to PEF pretreatments were determined using texture profile analysis and results are summarized in Table 3. The observed differences in texture parameters between air−dried and freeze−dried samples could be attributed to variations in cell structure and water removal kinetics during the drying process. Air drying leads to denser samples due to cell shrinkage, resulting in increased hardness compared to freeze−dried samples. However, the freeze−drying process might introduce structural changes that affect the ease of breaking, contributing to the observed differences in hardness. Air−dried samples were denser than freeze−dried samples. A previous study by Acevedo et al. [33] also reported that samples dried under vacuum at 50 °C showed higher hardness values than freeze−dried samples at similar relative humidity (RH).

PEF treatment, in combination with drying methods, further influenced the texture of dried apricots. Specifically, apricots that were treated with PEF and subsequently air dried had significantly (*p* < 0.05) higher values for hardness, gumminess and chewiness compared to those that were freeze dried (see Table 3). In air−dried apricots, the enhanced water removal due to increased cell membrane permeability could lead to a denser, harder texture as cells collapse more fully. In freeze−dried apricots, despite increased membrane permeability, the quick transition from frozen to dry minimizes the time for significant cellular collapse, hence maintaining a softer texture [34,35,36]. Previous research by Acevedo et al. [33] also reported that untreated control samples had denser shrinkage of cell walls compared to freeze−dried samples, resulting in a larger fracture force. In this study, dehydrated cell walls provided structural integrity and mechanical resistance to freeze−dried apple discs.

PEF pretreatment can significantly impact the texture parameters of dried apricots depending on the drying method and the intensity of PEF treatment influenced texture significantly. In air−dried apricot samples, high−intensity PEF pretreatments resulted in significantly (*p* < 0.05) higher hardness and gumminess values compared to low− and medium−intensity PEF−treated samples. For freeze−dried apricot samples, on the other hand, hardness values significantly (*p* < 0.05) increased after applying PEF pretreatments at low and high intensities compared to the control sample. This can be explained by changes in the microstructure of the dried apricots, as shown in Figure 1. Interestingly, hardness significantly (*p* < 0.05) decreased with medium−intensity PEF pretreatments compared to low− and high−intensity PEF pretreatments.

In terms of chewiness, there was a significant increase (*p* < 0.05) only in high−intensity PEF−treated samples for both drying methods compared to control. High−intensity PEF treatment can have more effect on disrupting these cell walls, causing structural changes that increase the resistance to deformation during chewing, resulting in a chewier texture [25]. Air−dried samples pretreated at medium PEF intensity had significantly (*p* < 0.05) the lowest adhesiveness compared to all other samples. However, resilience significantly (*p* < 0.05) decreased in PEF−treated air−dried samples compared to the control, with no significant differences between different PEF levels. This result suggests that PEF−treated air−dried apricot does not readily return to its original shape after being deformed during chewing. PEF treatment can enhance the mass transfer, which results in moisture distribution [25]. These changes may lead to a reduction in the sample’s ability to withstand deformation and recover to its original shape, resulting in decreased resilience. For freeze−dried apricots, chewiness and gumminess values significantly increased (*p* < 0.05) only when subjected to high−intensity PEF (1.8 kV/cm) compared to the control. Like air drying, higher PEF intensity can increase damage to cell structure, which may result in increased chewiness and gumminess. No significant changes in adhesiveness, resilience, cohesion, and springiness were found with different PEF pretreatments in the current study.

### 3.3. Scanning Electron Microscopy (SEM) Imaging and Analysis

Figure 1 shows the microstructure of dried apricot flesh and skin surface (A–H) after PEF pretreatments at low, medium, and high PEF intensities. During the drying process, water loss and segregation of components can result in rigidity of the cell walls. The extent of damage to the cellular walls will differ. Depending on the drying methods used, cellular wall collapse may also occur [37]. Overall, the PEF pretreatments and drying methods did not have much effect on the surface of apricot skin.

PEF pretreatments did not have a significant impact on the microstructure of air−dried apricot samples. Microstructure of air−dried apricot samples (Figure 1A–D, pulp) showed that the observed changes in structure were mainly attributed to the drying process. Specifically, the apricot flesh was tightly packed together and shrunk during the air−drying process. No noticeable differences were observed in the microstructure of different PEF−pretreated samples (Figure 1B–D, pulp) compared to the control sample (Figure 1A). Air drying caused the collapse of fruit structure. This collapse occurs due to significant moisture gradients within the product, leading to microstructural stresses that cause most of the capillaries to collapse and result in irreversible structural changes [38].

The microstructure of freeze−dried apricot samples (Figure 1E–H, pulp) showed that these samples exhibited better retention of shape, volume and honeycomb structure of apricot flesh compared to air drying. The structures of freeze−dried samples were similar, with more visible pores. This is likely due to the quick−freezing step in the freeze−drying process, which helps protect the food structure and shape, resulting in minimal volume reduction [39]. Apricot flesh with intact cytoplasmic membranes and cell walls naturally has a honeycomb structure [40]. The extent of structure collapse increased with increasing electric field strength in PEF pretreatment of the fruit (Figure 1E–H, pulp). Additionally, the cell wall structure collapsed more with increasing PEF strength, resulting in an increase in pore size. This finding is consistent with previous studies on PEF−pretreated freeze−dried apples, which also reported an increase in pore size with increasing PEF strength [12,41].

### 3.4. Chemical Analysis

#### 3.4.1. Antioxidant Activities

Antioxidant results using the FRAP and CUPRAC assays varied (Table 4). The antioxidant values using the CUPRAC assay were higher in dried apricots compared to the FRAP assay for all treatments. This could be due to differences in reactions of the different assays. FRAP assay produced lower antioxidant values than the CUPRAC assay [42]. This is because the FRAP method is not fast enough to oxidize certain thiol−type antioxidants. Additionally, the FRAP assay is conducted at an unrealistic acidic pH (pH = 3.6), whereas the CUPRAC assay is conducted at a physiological pH (pH = 7). In acidic conditions, the reducing capacity may be suppressed due to protonation of antioxidant compounds, which can further contribute to the lower antioxidant values obtained from the FRAP assay.

PEF pretreatments resulted in significantly higher antioxidant values in freeze−dried samples compared to air−dried samples using the FRAP assay. Previous study also reported a significant increase in antioxidant activity using the FRAP assay for PEF−treated apricots [7]. However, the CUPRAC assay showed the opposite trend with significantly lower antioxidant values in freeze−dried samples compared to air−dried samples with the same level of PEF pretreatment. This may be due to the heat produced during air drying that promotes the synthesis of new antioxidant compounds [43]. Papoutsis et al. [44] found that increasing the drying temperature resulted in higher antioxidant capacity of lemon samples, with air−dried and vacuum−dried samples showing significantly higher antioxidant values than freeze−dried samples.

High−intensity PEF treatment resulted in significantly higher antioxidant values compared to low−intensity PEF treatment and control samples using the CUPRAC assay. The intensity and duration of the PEF treatment play a significant role in the process. High−intensity treatments can cause greater cell disruption, potentially releasing more antioxidants or leading to oxidative stress, which may degrade antioxidants [9]. Variations in frequency and pulse width can affect the amount of stress applied to the fruit cells, and the different settings can result in varying impacts on antioxidant levels (Timmermans et al., 2019). Previous studies have reported that increasing PEF intensity can enhance antioxidant activity of blackcurrant [7,45,46]. In addition, all PEF−pretreated samples that were freeze dried showed a significant increase (*p* < 0.05) in antioxidant values using the FRAP assay compared to control samples. Vuong et al. [47] found that the antioxidant values of freeze−dried chasteberry (Vitex agnus−castus) leaves using both the FRAP and CUPRAC methods were significantly higher than those of oven−dried samples.

#### 3.4.2. Total Phenol Content (TPC)

Table 4 summarizes the TPC results of PEF−pretreated dried apricots, with values ranging from 0.445 to 0.527 mg GAE/g for air−dried samples and 0.466 to 0.576 mg GAE/g for freeze−dried samples. Only high−PEF pretreatments significantly (*p* < 0.05) increased the TPC of air−dried samples compared to control and low−PEF−treated samples. PEF treatment can help improve the extraction of TPC from date palm fruits due to the permeabilisation of the cell membrane, which can enhance mass transfer [48]. The permeabilisation of plant cells caused by PEF treatments can enhance the intracellular content extraction of various plant products, including orange juice [49], almond extract [50] and date juice [51]. However, for freeze−dried samples, PEF pretreatments resulted in no significant differences in TPC compared to the control samples.

#### 3.4.3. Metabolite Profiling

Heat maps showing the mean intensities of different metabolites as a function of the effect of PEF pretreatment on apricots that were subsequently air or freeze dried are presented in Figure 2. In Figure 2a, two distinct clusters of fatty acids differentiated air− and freeze−dried samples. The freeze−dried samples had significantly higher levels of fatty acids compared to the air−dried samples. Freeze drying is more effective in preserving the nutritional content and functional components of fruits compared to air drying [52]. Freeze drying retains more fatty acids in fruits compared to air drying. A study on lemon juice vesicles reported a loss of 67.68% in total fatty acids after air drying, 53.00% in conventional freeze drying and 36.95% in integrated freeze drying [53]. Additionally, most fatty acid levels increased with increasing PEF intensity. This could be attributed to enhanced cell permeabilization with increasing PEF intensity, which promotes the release of bioactive components in plant tissues [9]. However, there are limited studies that specifically focus on the changes in fatty acid content of PEF−pretreated fruit. PEF pretreatment of grape juice (35 kV/cm, bipolar 4 μs square pulses at 1000 Hz for 1 μs) did not result in a significant change in total fatty acids compared to the control group [23]. However, in terms of free fatty acids, no significant changes were observed except for lauric acid that decreased compared to the control group.

According to the findings presented in Figure 2b, the drying techniques used resulted in two major clusters of amino acids. The amino acids in cluster I were more abundant in freeze−dried samples, while the amino acids in cluster II were higher in air−dried samples. In freeze−dried samples, PEF pretreatment led to significantly higher levels of amino acids compared to the control group. PEF technology has been shown to influence the amino acid content of fruits and vegetables [23,54]. Zhao et al. [54] reported a significant (*p* < 0.05) increase in total free amino acids, especially theanine, in green tea treated with PEF (20–40 kV/cm, 200 μs). Likewise, an increase (*p* < 0.05) in individual amino acids such as histidine, tryptophan, asparagine, phenylalanine and ornithine in grape juice after PEF treatment was observed [23].

In cluster I shown in Figure 2c, sugar content in air−dried samples showed significantly (*p* < 0.05) higher sugar content than freeze−dried samples. This was confirmed by a study on lemon juice vesicles, which reported that air drying resulted in the highest total soluble sugars (17.12 ± 0.20 mg/g), which were 1.24 and 1.49 mg/g higher than those of integrated freeze drying (IFD) and conventional freeze drying (CFD), respectively [55]. These findings suggest that air drying may better preserve the sugar content in some fruits.

Figure 2d further differentiated dried apricot samples in terms of organic acid content based on the drying methods. Cluster I exhibited higher levels of organic acids in air−dried samples, while cluster II showed higher levels of organic acids in freeze−dried samples. This highlights the different impacts of freeze drying and air drying on the organic acid composition of fruits. A study on European cranberry found that lyophilized fruit had significantly higher levels of organic acids compared to fruit dried at temperatures of 35–40 °C [56]. The largest relative difference was observed in the case of ascorbic acid, with the content in the high−temperature−dried fruit averaging 5 mg/100 g of dry matter (DM), representing a 42% decrease compared to the lyophilised material. Other organic acids (quinic, citric and malic acids) exhibited an average difference of 0.9–2.1 g/100 g DM (14–17%). On the other hand, in the case of air drying, drying of gilaburu fruits at different temperatures (50, 60 and 70 °C) resulted in a significant loss of organic acids and other bioactive components, with the highest loss observed at 70 °C [57].

Sparse partial least squares–discriminant analysis (sPLS−DA) was applied to summarize the metabolite differences between the control and the PEF−pretreated (at low, medium, and high PEF intensities) apricot samples that were subjected to either air drying or freeze drying. Figure 3a,b illustrate that four components were sufficient to explain the dataset, with a 6.2% error rate in classification of the samples.

In Figure 4a, the first two latent variables explained 32.4% of the variation in the X data and 15.9% of the variation in the Y data. For component 1, the air−dried sample had a high positive score and was separated from the freeze−dried sample that had a high negative score. Furthermore, PEF−treated freeze−dried samples, especially those treated at high PEF intensities, had high negative scores for component 2 and were further separated from control freeze−dried samples that had a high positive score.

The loadings depicted in Figure 5a, and the ANOVA results (Appendix A) showed significant changes (*p* < 0.05) in three amino acids (threonine, trans−4−hydroxyproline and histidine) and a tripeptide (glutathione) depending on the method of drying. Threonine and hydroxyproline levels were significantly higher in air−dried samples, while the other amino acids were significantly higher in freeze−dried samples. Fatty acids (palmitic acid, stearic acid and arachidic acid) were all significantly higher in freeze−dried samples. Among the organic acids, only coumaric acid, 2−ketoglutamaric acid and phosphoric acid were significantly higher in freeze−dried samples, while the other organic acids were all significantly higher in air−dried samples. As for sugars, glucopyranose, galactopyranose and fructose were all significantly higher in air−dried samples. Similarly, Xie et al. [55] reported that air drying led to a significantly higher total amino acid content compared to integrated freeze drying (IFD) and conventional freeze drying (CFD). In addition, CFD resulted in the notably highest total organic acid content, while air drying (AD) contributed to the notably highest total soluble sugar content.

The loadings depicted in Figure 5b and the ANOVA results (Appendix A) showed significant changes in metabolites in the freeze−dried, high−PEF−treated sample compared to the non−PEF−treated freeze−dried control sample. All amino acids (glycine, leucine, norleucine, threonine, asparagine, cis−4−hydroxyproline, phenylalanine and lysine) were significantly higher in the high−intensity PEF−treated freeze−dried sample compared with the non−PEF freeze−dried control sample. Previous study reported that the total free amino acids abundant in PEF−treated alcoholic beverages made of date palm fruits were significantly (*p* < 0.05) higher than in the untreated control. Increasing PEF intensity (2.92 > 2.02 > 1.38 kV/cm) led to a corresponding increase in the production of free amino acids [23]. In addition, essential amino acids (arginine, histidine, leucine, isoleucine, lysine, phenylalanine and threonine) were also significantly (*p* < 0.05) higher in apricot samples treated with PEF at higher intensity (2.92 kV/cm) compared to control.

Myristic acid and octadecatrienoic acid, both fatty acids, were significantly higher in the high−intensity−PEF freeze−dried sample. Among the organic acids, D−citramalic acid, malic acid, isocitric acid, ribonic acid and shikimic acid were significantly higher in the high−intensity−PEF freeze−dried sample compared to the non−PEF freeze−dried control sample. However, sedoheptulose and D−allose sugars were significantly lower in high−intensity PEF−treated freeze−dried samples. Previous studies have shown that PEF treatment had no significant effect on saturated fatty acids and mono− and polyunsaturated fatty acids and only caused a slight decrease in total fatty acid content [23,58,59].

In Figure 4b, the first two latent variables explained 9.4% of the variation in the X data and 7.5% of the variation in the Y data. For component 3, the non−PEF−treated air−dried control sample had a high positive score and was distinguished from the medium−PEF−treated air−dried sample with a high negative score. The control freeze−dried sample, with a high negative score, was further separated from low−PEF−treated freeze−dried sample that had a high positive score for component 4.

The loadings depicted in Figure 5c and the results from ANOVA (Appendix A) showed a significant change in metabolites in the air−dried, medium−PEF−treated sample compared to the non−PEF−treated air−dried control sample. Alanine, aspartic acid, and proline, among the amino acids, were significantly higher in the medium−PEF−treated air−dried sample compared with the non−PEF−treated air−dried control sample. The mechanism by which PEF increases amino acids is not fully understood, but it is thought to involve disruption of organelles and vacuoles, allowing proteases to access and degrade proteins [23], as well as changes in the ζ−potential of the cell membrane, leading to the degradation of peptides [60,61].

Among the organic acids, fumaric acid, benzoic acid, methylpropionic acid, malic acid, citramalic acid, isocitric acid, ferulic acid, 2−butenedioic acid, itaconic acid, pyrrolidine carboxylic acid, threonic acid and trans−4−(aminomethyl)cyclohexanecarboxylic acid were significantly lower in medium−PEF−treated air−dried samples compared with the non−PEF−treated air−dried control samples. Gentiobiose, one of the sugars, was significantly higher in the medium−PEF−treated air−dried sample compared with the control.

The loadings depicted in Figure 5d and the results from ANOVA (Appendix A) showed a significant change in metabolites caused by low−intensity PEF treatment in freeze−dried samples compared with the non−PEF−treated freeze−dried control samples. Hydroxyproline, an amino acid, was significantly higher in the low−PEF−treated freeze−dried sample compared with the control. Among the organic acids, malonic acid, oxaloacetic acid, citraconic acid, cis−aconitic acid, citric acid, ferulic acid, propanedioic acid, hydracrylic acid, galactonic acid and methylsuccinic acid were significantly lower in low−intensity PEF−treated freeze−dried samples compared with non−PEF−treated freeze−dried control samples. The sugars D−xylopyranose, beta−L−galactopyranoside, sedoheptulose, L−threitol, D−(+)−trehalose, D−allose, glucose, rhamnose and ribose were all significantly lower in low−intensity PEF−treated freeze−dried samples except for glucose, which was significantly higher in the low−PEF−treated samples compared with the control.

Overall, the choice of drying method (air drying vs. freeze drying) significantly influenced the metabolite composition of apricot samples. Air drying tended to preserve certain amino acids and sugars better, while freeze drying resulted in higher levels of fatty acids and specific organic acids. Additionally, PEF treatment exerted further significant effects, particularly in combination with freeze drying, suggesting its potential for modulating the metabolite composition of dried apricot samples.

## 4. Conclusions

The study aimed to investigate the impact of pulsed electric field (PEF) treatment on the physicochemical properties of air−dried and freeze−dried apricots. The results indicated that PEF treatment had no significant effect on the moisture content or water activity of the dried fruits. However, it did have a notable influence on the texture, antioxidant activity and bioactive compounds of the dried apricots. The most effective method for preserving apricots appeared to be freeze drying combined with high−intensity PEF pretreatments. This approach not only improved texture but also preserved the antioxidant activity and bioactive compounds, enhancing both consumer appeal and health benefits.

Air drying with high−intensity PEF resulted in increased firmness, gumminess, and chewiness, potentially leading to a denser texture that may be less appealing. On the other hand, freeze drying resulted in improved texture. Freeze−dried apricots treated with high−intensity PEF (1.8 kV/cm) not only resulted in desirable texture but also exhibited significant increases in antioxidant activity, amino acids, and fatty acids. Future research could further investigate the different PEF intensities and their specific effects on the sensory properties of freeze−dried apricots. In this way, the research can further provide valuable information for optimizing the drying process in relation to the sensory attributes and consumer perception of dried apricots.

## Figures and Tables

**Figure 1 foods-13-01764-f001:**
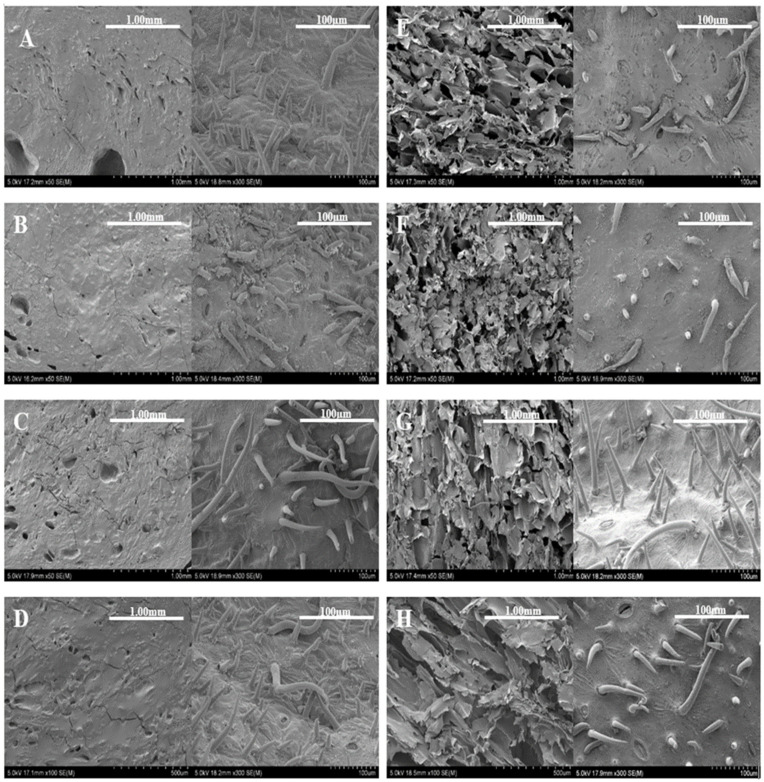
Scanning electron microscopy of the pulp structure (**left**) and surface of apricot skin (**right**) after PEF pretreatment and drying processes. (**A**–**H**): Different drying and PEF pretreatment conditions: (**A**) air drying, control, (**B**) air drying, low−intensity PEF, (**C**) air drying, medium−intensity PEF, (**D**) air drying, high−intensity PEF; (**E**) freeze drying, control, (**F**) freeze drying, low−intensity PEF, (**G**) freeze drying, medium−intensity PEF, (**H**) freeze drying, high−intensity PEF. Low−intensity PEF: 0.7 kV/cm, 50 Hz every 20 µs for 30 s; Medium−intensity PEF: 1.2 kV/cm, 50 Hz every 20 µs for 30 s; High−intensity PEF: 1.8 kV/cm, 50 Hz every 20 µs for 30 s.

**Figure 2 foods-13-01764-f002:**
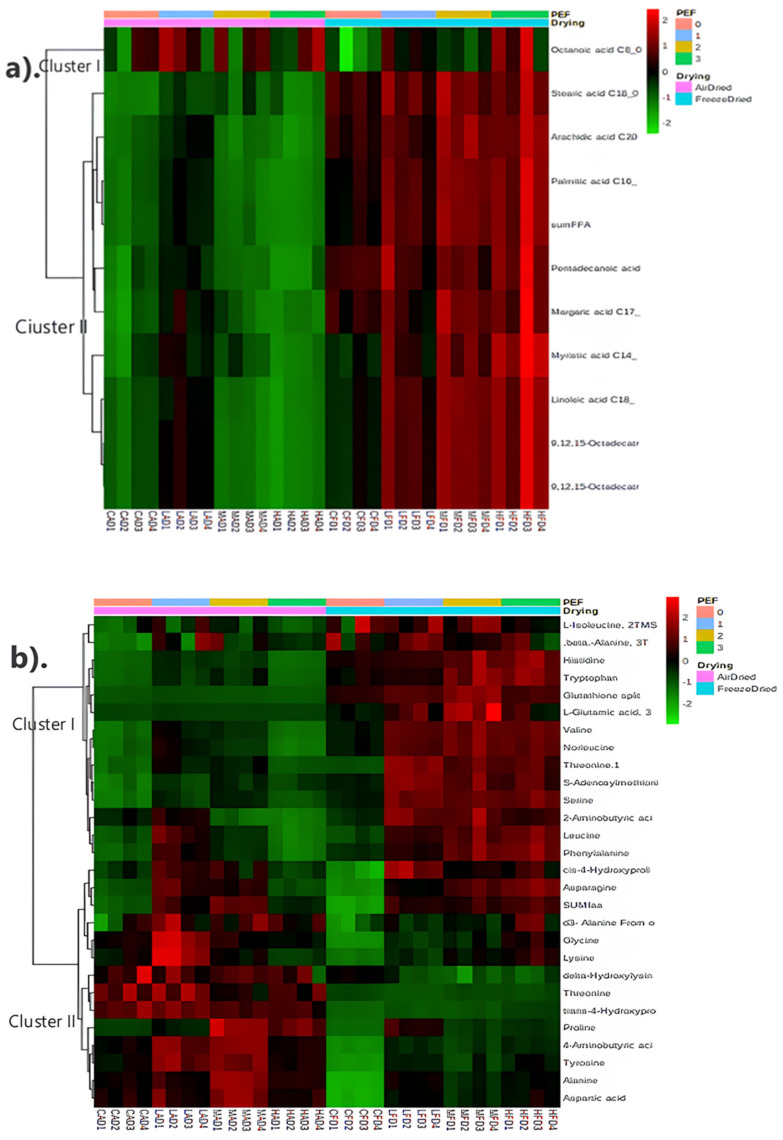
Heat maps showing the intensity of different metabolites in PEF−treated and non−PEF−treated dried apricots: (**a**) Fatty acids, (**b**) Amino acids, (**c**) Sugars and (**d**) Organic acids. Shades of green to red represent increasing intensity of the metabolites. The first row of each heat map indicates the intensity of PEF treatment: low (1), medium (2), high (3) and control (0).

**Figure 3 foods-13-01764-f003:**
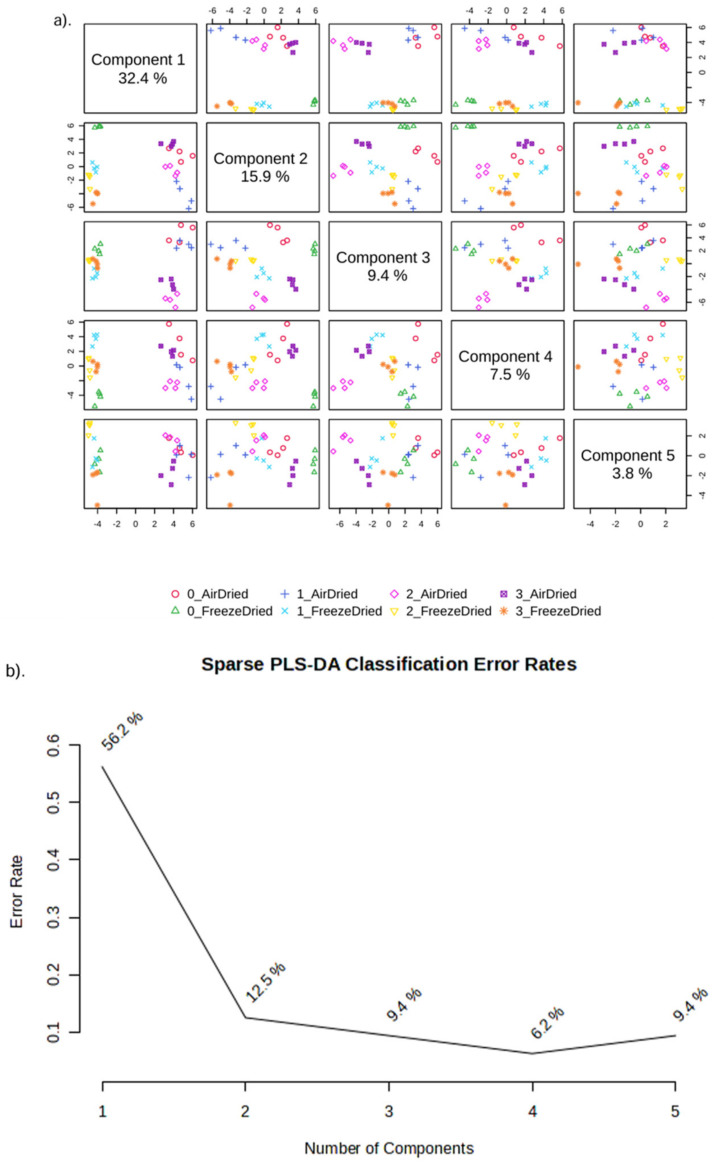
sPLS−−DA model showing: (**a**) Overview of score plots of dried apricot samples preprocessed with different PEF treatments as explained by five components of the selected metabolites; (**b**) Performance of the model evaluated using cross−validation. Numbers 0, 1, 2 and 3 indicate (i) No PEF, (ii) Low PEF (0.7 kV/cm), (iii) Medium PEF (1.2 kV/cm) and (iv) High PEF (1.8 kV/cm) treatments, respectively.

**Figure 4 foods-13-01764-f004:**
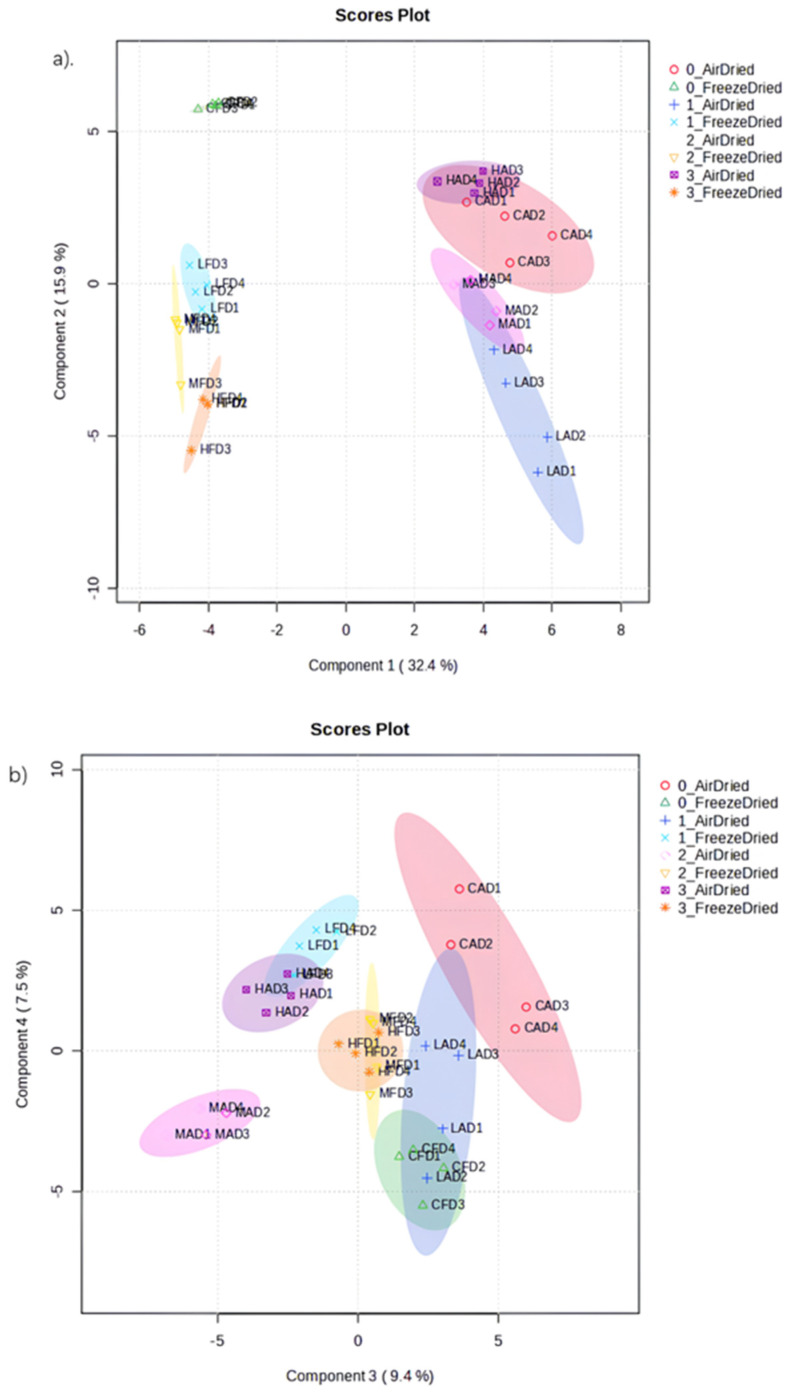
Two−dimensional (2−D) score plots showing dried apricot samples pretreated under different PEF conditions as explained by the components: (**a**) 1 and 2; and (**b**) 3 and 4 using the sPLS−DA model with selected metabolites: Numbers 0, 1, 2 and 3 indicate (i) No PEF, (ii) Low PEF (0.7 kV/cm), (iii) Medium PEF (1.2 kV/cm) and (iv) High PEF (1.8 kV/cm) treatments, respectively.

**Figure 5 foods-13-01764-f005:**
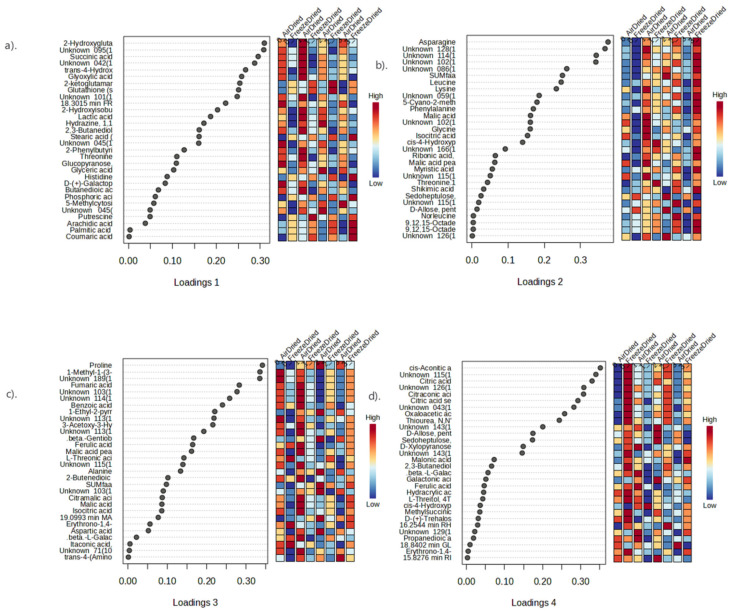
Loading plots of the top−ranked features. The loadings plot shows the variables selected by the sPLS−DA model for a given component. The variables are ranked by the absolute values of their loadings. Where: **a**, **b**, **c** and **d** correspond to loadings for components 1, 2, 3 and 4, respectively.

**Table 1 foods-13-01764-t001:** Pulse electric field pre-treatments of apricot samples dried using either air-drying or freeze-drying.

No	ID	PEF	Drying	Replicate
1	C−AD1	No PEF	Air drying	1
2	C−AD2	No PEF	Air drying	2
3	C−AD3	No PEF	Air drying	3
4	L−AD1	Low PEF	Air drying	1
5	L−AD2	Low PEF	Air drying	2
6	L−AD3	Low PEF	Air drying	3
7	M−AD1	Medium PEF	Air drying	1
8	M−AD2	Medium PEF	Air drying	2
9	M−AD3	Medium PEF	Air drying	3
10	H−AD1	High PEF	Air drying	1
11	H−AD2	High PEF	Air drying	2
12	H−AD3	High PEF	Air drying	3
13	FDC1	No PEF	Freeze drying	1
14	FDC2	No PEF	Freeze drying	2
15	FDC3	No PEF	Freeze drying	3
16	FDL1	Low PEF	Freeze drying	1
17	FDL2	Low PEF	Freeze drying	2
18	FDL3	Low PEF	Freeze drying	3
19	FDM1	Medium PEF	Freeze drying	1
20	FDM2	Medium PEF	Freeze drying	2
21	FDM3	Medium PEF	Freeze drying	3
22	FDH1	High PEF	Freeze drying	1
23	FDH2	High PEF	Freeze drying	2
24	FDH3	High PEF	Freeze drying	3

C, L, M and H indicate control, low PEF (0.7 kV/cm), medium PEF (1.2 kV/cm), and high PEF (1.8 kV/cm). AD and FD indicate air drying and freeze drying, respectively.

**Table 2 foods-13-01764-t002:** Moisture content and water activity of different PEF−pretreated apricots at either 0.7, 1.6 or kV/cm for 50 hz every 20 µs for 30 s that were then subjected to air drying or freeze drying.

	PEF Intensity	Moisture Content	Aw
Air drying			
1	Control	27.256 ± 1.868 ^Aa^	0.544 ± 0.009 ^Aa^
2	Low PEF	27.231 ± 0.869 ^Aa^	0.559 ± 0.014 ^Aa^
3	Medium PEF	27.47 ± 0.507 ^Aa^	0.57 ± 0.014 ^Aa^
4	High PEF	27.354 ± 0.943 ^Aa^	0.571 ± 0.016 ^Aa^
*p* value (PEF condition)	Pr > F	*p* > 0.05	*p* > 0.05
*p* value (drying method)	Pr > F	**	***
Freeze drying			
1	Control	19.716 ± 0.460 ^Ab^	0.384 ± 0.010 ^Ab^
2	Low PEF	19.853 ± 0.302 ^Ab^	0.399 ± 0.024 ^Ab^
3	Medium PEF	19.546 ± 0.165 ^Ab^	0.392 ± 0.010 ^Ab^
4	High PEF	19.42 ± 0.865 ^Ab^	0.399 ± 0.017 ^Ab^
*p* value (PEF condition)	Pr > F	*p* > 0.05	*p* > 0.05

(^A^) describe the significant differences between different PEF treatments. (^a, b^) describe the significant differences between different drying methods. (**, ***) indicate (*p* < 0.01 and *p* < 0.001), respectively.

**Table 3 foods-13-01764-t003:** Texture of different PEF−pretreated apricots at either 0.7, 1.6 or kV/cm for 50 hz every 20 µs for 30 s that were then subjected to air drying or freeze drying.

	PEF Intensity	Texture						
		Hardness	Adhesiveness	Resilience	Cohesion	Springiness	Gumminess	Chewiness
Air drying								
1	Control	137.153 ± 23.637 ^Ca^	0.032 ± 0.187 ^Aa^	35.844 ± 4.027 ^Aa^	0.931 ± 0.140 ^Aa^	190.442 ± 177.722 ^Aa^	126.961 ± 22.028 ^Ca^	254.287 ± 253.539 ^Ba^
2	Low PEF	716.382 ± 111.191 ^Ba^	−1.108 ± 0.824 ^Aa^	29.005 ± 0.377 ^Ba^	0.778 ± 0.010 ^Aa^	90.716 ± 2.023 ^Aa^	556.648 ± 78.866 ^Ba^	505.374 ± 76.017 ^ABa^
3	Medium PEF	827.646 ± 67.560 ^Ba^	−9.289 ± 3.96 ^Bb^	24.788 ± 1.093 ^Ba^	0.753 ± 0.020 ^Aa^	92.709 ± 2.226 ^Aa^	624.145 ± 67.131 ^Ba^	579.48 ± 74.965 ^ABa^
4	High PEF	1052.633 ± 11.842 ^Aa^	3.673 ± 1.681 ^Aa^	25.715 ± 2.754 ^Ba^	0.743 ± 0.056 ^Aa^	93.798 ± 11.797 ^Aa^	781.163 ± 50.504 ^Aa^	731.158 ± 83.063 ^Aa^
*p* value (PEF condition)	Pr > F	***	*	*	*p* > 0.05	*p* > 0.05	***	*
*p* value (drying method)	Pr > F	*p* > 0.05	*p* > 0.05	*p* > 0.05	*p* > 0.05	*p* > 0.05	*p* > 0.05	*p* > 0.05
Freeze drying								
1	Control	237.689 ± 62.479 ^Ca^	−0.958 ± 2.972 ^Aa^	25.467 ± 5.551 ^Aa^	0.709 ± 0.041 ^Aa^	82.639 ± 14.783 ^Aa^	167.448 ± 37.869 ^Ba^	139.356 ± 43.717 ^Ba^
2	Low PEF	342.77 ± 12.242 ^Bb^	−0.016 ± 0.675 ^Aa^	20.134 ± 2.539 ^Ab^	0.598 ± 0.068 ^Ab^	64.274 ± 2.671 ^Ab^	205.134 ± 28.058 ^Bb^	132.34 ± 23.158 ^Bb^
3	Medium PEF	209.267 ± 2.109 ^Cb^	−0.939 ± 0.695 ^Aa^	19.219 ± 1.541 ^Ab^	0.634 ± 0.166 ^Aa^	75.807 ± 20.476 ^Aa^	132.583 ± 34.2 ^Bb^	104.974 ± 51.574 ^Bb^
4	High PEF	472.137 ± 27.969 ^Ab^	−0.329 ± 0.923 ^Ab^	25.048 ± 2.55 ^Aa^	0.695 ± 0.012 ^Aa^	73.038 ± 2.854 ^Ab^	328.234 ± 20.657 ^Ab^	240.129 ± 24.603 ^Ab^
*p* value (PEF condition)	Pr > F	*p* > 0.05	***	*p* > 0.05	*p* > 0.05	*p* > 0.05	**	*

(^A, B, C^) describe the significant differences between different PEF treatments. (^a, b^) describe the significant differences between different drying methods. (*, **, ***) indicate (*p* < 0.05, *p* < 0.01, *p* < 0.001), respectively.

**Table 4 foods-13-01764-t004:** Antioxidant activity and TPC of different PEF−pretreated apricots at either 0.7, 1.6 or kV/cm for 50 hz every 20 µs for 30 s that were then subjected to air drying or freeze drying.

	PEF Intensity	Antioxidant Ability		Total Polyphenol Content
		mg of Trolox Equivalent/g Dried Powder (FRAP)	mg of Trolox Equivalent/g Dried Powder (CUPRAC)	mg of Gallic Acid Equivalent/g Dried Powder
Air drying				
1	Control	0.447 ± 0.016 ^Bb^	4.742 ± 0.27 ^Ca^	0.445 ± 0.015 ^Ba^
2	Low PEF	0.565 ± 0.021 ^Ab^	6.626 ± 0.174 ^Ba^	0.466 ± 0.016 ^Ba^
3	Medium PEF	0.567 ± 0.016 ^Ab^	7.031 ± 0.115 ^ABa^	0.493 ± 0.030 ^ABa^
4	High PEF	0.607 ± 0.02 ^Ab^	7.417 ± 0.129 ^Aa^	0.527 ± 0.020 ^Aa^
*p* value (PEF condition)	Pr > F	*	*	*
*p* value (drying method)	Pr > F	*p* > 0.05	*p* > 0.05	*p* > 0.05
Freeze drying				
1	Control	0.936 ± 0.111 ^Ba^	4.363 ± 0.498 ^Aa^	0.576 ± 0.139 ^Aa^
2	Low PEF	1.458 ± 0.161 ^Aa^	4.617 ± 0.154 ^Ab^	0.503 ± 0.024 ^Aa^
3	Medium PEF	1.315 ± 0.074 ^Aa^	4.062 ± 0.353 ^Ab^	0.466 ± 0.014 ^Aa^
4	High PEF	1.479 ± 0.048 ^Aa^	5.963 ± 0.061 ^Ab^	0.522 ± 0.006 ^Aa^
*p* value (PEF condition)	Pr > F	*	*p* > 0.05	*p* > 0.05

(^A, B, C^) describe the significant differences between different PEF treatments. (^a, b^) describe the significant differences between different drying methods. (*) indicate *p* < 0.05.

## Data Availability

The original contributions presented in the study are included in the article, further inquiries can be directed to the corresponding author.

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
