# Peer review of "Pulsed Electric Field Pretreatments Affect the Metabolite Profile and Antioxidant Activities of Freeze− and Air−Dried New Zealand Apricots"

_foods, 2024, doi:10.3390/foods13111764_

Round 1

Reviewer 1 Report

Comments and Suggestions for Authors

This article has explored the use of PEF on different attributes of dried apricots. Different intensities of the electric field were tested and 2 drying methods were employed.

Results are interesting but the data analysis and the discussion of the results should be improved.

Please be careful with some typos and small grammatical mistakes throughout the text.

Please also check that your citations in the text follow the journal’s rules, see for example, page 2, lines 49-52.

No need for Table 1. It is enough that, in the text, you report the average plus/minus the std for each measurement.

Page 4, lines 147-165. Report the pulse width, the number of pulses and the pulses frequency. Some info is repeated in the text. What is the timing between the PEF treatment and the drying procedure?

No need of the column “Cultivar” in Table 2. Already written on Page 3, line 141.

“3.2 Texture”. There is practically no discussion in this section. Not clear the connection you want to establish between increased moisture diffusivity and PEF-modified texture.

Legend of Figure 1. Be specific on what is shown in the right figure and what in the left figure of A-H.

“3.4.2” Not clear the connection you want to establish between enhanced extraction of plant products and higher TPC in the PEF-treated samples

The metabolite analysis in the results section is neither very informative nor reader friendly. I suggest you work a bit more to have a better way to show the data of the multivariate analysis. As a first suggestion, you should try to use the orthogonal projections to latent structures (OPLS) method instead of the PLS-DA method. In this way you will have more reliable and probably also more clear results of the multivariate analysis. Then you could use S-plots to identify compounds most influential to the separation of the groups in the OPLS model and, finally use ANOVA to analyse the selected compounds. Please see Wiklund, S., et al (2008). Visualization of GC/TOF-MS-based metabolomics data for identification of biochemically interesting compounds using OPLS class models. Analytical Chemistry, 80, 115–122.

“The higher the PEF intensity, the more metabolites produced”. A bit ambiguous, the S-plots suggested above may allow you to identify the most interesting compounds as well as potential biomarkers.

The conclusion part is phrased like an abstract more than conclusions as such.

Comments on the Quality of English Language

some typos and small grammatical mistakes throughout the text.

Author Response

Reviewer 1

This article has explored the use of PEF on different attributes of dried apricots. Different intensities of the electric field were tested and 2 drying methods were employed.

Results are interesting but the data analysis and the discussion of the results should be improved.

Please be careful with some typos and small grammatical mistakes throughout the text.

Please also check that your citations in the text follow the journal’s rules, see for example, page 2, lines 49-52.

Response: The citations in text has been amended following the Journal’s rules. See for example, Page1 line 32

No need for Table 1. It is enough that, in the text, you report the average plus/minus the std for each measurement.

Response: Table deleted page4, line 168 

Page 4, lines 147-165. Report the pulse width, the number of pulses and the pulses frequency. Some info is repeated in the text.

Response: The repeated text has been deleted at page4, line 159-160 which reads,

Four different field strength levels of PEF pretreatments were carried out in this study: i) No PEF, ii) Low PEF (0.7kV/cm), iii) Medium PEF (1.2kV/cm) and iv) High PEF (1.8kV/cm), with a frequency of 50 Hz, and electric pulses coming in every 20 µs for 30 s.”

What is the timing between the PEF treatment and the drying procedure?

Response: For air drying, samples are dried directly after PEF. For freeze-drying, samples were freeze froe 12 hours before freeze-drying. This detail has been added to the page4, line 160-164 which reads

One half of the apricot samples were subjected to air drying and the other half to freeze drying. Directly after PEF treatment, the apricot samples were air dried at 70°C for 12 h. For freeze drying, the apricot samples were frozen first using a Multifresh blast chiller (Irinox, USA) for 12 hours at -20 °C. Then the samples were moved to a VirTis 35L general purpose freeze dryer (SP Scientific, USA).”

No need of the column “Cultivar” in Table 2. Already written on Page 3, line 141.

Response: The column “Cultivar” has been deleted at page4, line 169

“3.2 Texture”. There is practically no discussion in this section.

Response: The whole paragraph has been rewrote page13, line 337-386

Not clear the connection you want to establish between increased moisture diffusivity and PEF-modified texture.

Response: The moisture diffusivity has been deleted at page 13 line 352-254, which reads,

“This suggests that PEF treatment can enhance the drying of fruits by improving the permeabilization of cell membranes (Amami, 2008; Amami, 2005; Wiktor, 2014).”

Legend of Figure 1. Be specific on what is shown in the right figure and what in the left figure of A-H.

Response: Pulp structure (left) and surface of apricot skin (right) This detail has been added to the page15, line 416 which reads,

Scanning electron microscopy of the pulp structure (left) and surface of apricot skin (right)”

“3.4.2” Not clear the connection you want to establish between enhanced extraction of plant products and higher TPC in the PEF-treated samples

Response:The connection hear is that PEF treatment could help improved the extraction of TPC due to the permeabilization which enhance the mass transfer

Page16, line 463-365 which reads,

PEF treatment can help improve the extraction of TPC from date palm fruits due to the permeabilization of cell walls, which can enhance mass transfer.”

The metabolite analysis in the results section is neither very informative nor reader friendly. I suggest you work a bit more to have a better way to show the data of the multivariate analysis. As a first suggestion, you should try to use the orthogonal projections to latent structures (OPLS) method instead of the PLS-DA method. In this way you will have more reliable and probably also more clear results of the multivariate analysis. Then you could use S-plots to identify compounds most influential to the separation of the groups in the OPLS model and, finally use ANOVA to analyse the selected compounds. Please see Wiklund, S., et al (2008). Visualization of GC/TOF-MS-based metabolomics data for identification of biochemically interesting compounds using OPLS class models. Analytical Chemistry, 80, 115–122.

“The higher the PEF intensity, the more metabolites produced”. A bit ambiguous, the S-plots suggested above may allow you to identify the most interesting compounds as well as potential biomarkers.

Response: This section has been re analysis using sPLS-DA Page20-24, line 531-643

The reason why using sPLS-DA as following:

sPLDA is commonly used to ID sparse subset of variables that are discriminating between the groups similar to OPLSDA. However, sPLSDA adds sparsity by penalty addition to the PLS coefficients to encourage the non discriminating molecules to be zero. sPLSDA is leveraged in this study rather than OPLSDA as we assume that the discriminating molecules would be unique from one to another while taking the assumption of low multicollinearity in our dataset.

The conclusion part is phrased like an abstract more than conclusions as such.

Response: The conclusion part has been rewritten Page24, line 647-663

Reviewer 2 Report

Comments and Suggestions for Authors

The manuscript submitted for review: "Pulsed electric field pretreatments affect the metabolite profile and antioxidant activities of freeze- and air-dried New Zealand apricots" is a very interesting paper with many novel results. The manuscript is quite well written, nevertheless please consider the comments below.

Abstract
The most important qualitative results obtained in the study are missing from the abstract.
Line 10: "...to improve the quality...". - please indicate which quality parameters you are referring to
Line 24: which bioactive ingredients are involved?
The abstract lacks a conclusion on the electric field parameters used - which parameters resulted in the most beneficial characteristics of the apricots.

Indroduction
Line 35-36: what specific vitamins and minerals are found in apricots?
Line 67-80: please justify why these relationships occur.
Line119-130: why are amino acids and sugar called metabolites? What is the reason for the relationships described in this paragraph? Is it related to the higher extraction efficiency or to the lower water content of the products?

Material and methods
Line 148-159: why were these PEF parameters used? Were they determined on the basis of literature data or own experience?
Line 162-165: How long did the freeze-drying process take? What was the vacuum and temperature of the sublimation process?
Line 190-193: was the water content of the apricots determined on the basis of fruit subjected to sublimation or air drying?

Results and discussion
Line 304: instead of less should be higher.
Line 311-315: incorrectly described results for moisture content and aw for sublimation and air-dried samples.
Table 3 is very difficult to read. Please describe column 1 correctly. Please separate the results into 3 tables: 1. moisture and aw content; 2. texture characteristics; 3. antioxidant properties and polyphenol content.
Please correct the numbering of the subsections.
Please describe this section in more detail: "However, for all the texture parameters, there were no significant differences between air-dried and freeze-dried control samples". Differences did occur. There should be no bracket when quoting: Acevedo, Briones, Buera, & Aguilera.
Antioxidant properties: are the higher values after PEF treatment due to an increase in extraction efficiency or the formation of new compounds with higher antioxidant potential? Or rearrangements of existing compounds?
Metabolite profiling: figures 2 3 and are not very clear, they are unreadable. Please move the description of figure 2 below or above the figure.
Table 4: very difficult to read. Please separate the results into 3 tables: 1. amino acid content; 2. fatty acid content; 3. organic acid content.
Is it not possible to convert the amino acids to present them in other units in the table?
Please state from what you derive the relationships described in the paragraph: "In cluster 1 shown in Figure 3c, sugar content in air dried sample showed significantly p<0.05) higher sugar content than freeze dried sample".

Conclusion
Please write, based on the results obtained, which treatment could be recommended (both PEF and drying method) for the most beneficial preservation of apricots.

Author Response

Reviewer two

The manuscript submitted for review: "Pulsed electric field pretreatments affect the metabolite profile and antioxidant activities of freeze- and air-dried New Zealand apricots" is a very interesting paper with many novel results. The manuscript is quite well written, nevertheless please consider the comments below.

Abstract
The most important qualitative results obtained in the study are missing from the abstract.

Response: The missing physical results has been added to page1, line 16-19,which reads

“PEF pretreatments significantly (p<0.05) increased firmness of all the air-dried sample and most freeze-dried apricot samples compared to the control group. However, PEF treatment at 1.2kV/cm did not have any effect on hardness of the freeze-dried sample. The moisture content and water activity of freeze-dried samples were found to be significantly lower than those of air-dried samples.“

Line 10: "...to improve the quality...". - please indicate which quality parameters you are referring to

Response: In terms antioxidant activity and bioactive compounds. This detail has been added to page1, line11,which reads,

“Pulsed electric field (PEF) pretreatment has been shown to improve the quality of dried fruits in terms antioxidant activity and bioactive compounds. “

Line 24: which bioactive ingredients are involved?

Response: amino acids, fatty acids, sugar, organic acids and phenolic compounds. This detail has been added to Page1, line27-29 which reads.

The results of this study suggest that PEF pretreatment can influence the quality of air dried and freeze-dried apricots in terms antioxidant activity and metabolites such as amino acids, fatty acids, sugar, organic acids and phenolic compounds.”

The abstract lacks a conclusion on the electric field parameters used - which parameters resulted in the most beneficial characteristics of the apricots.

The most beneficial treatment for preserving apricots is freeze-drying combined with high-intensity(1.8kv/cm). This detail has been added to Page1, line29-31 which reads.

The most effective treatment for preserving the quality of dried apricots is freeze-drying combined with high-intensity(1.8kv/cm) PEF treatment.”

Indroduction
Line 35-36: what specific vitamins and minerals are found in apricots?

Response: VA,VC,VE and VK Potassium, Calcium, Iron and Magnesium. This detail has been added to the Page1, line41-42, which reads.

Apricots are rich in vitamins A, C, E, and K, carotenoids, polyphenols, and minerals such as potassium, calcium, iron, and magnesium (Ali et al., 2011).”

Line 67-80: please justify why these relationships occur.

Response: This may due to different among fruits. PEF treatment impacts cell membrane permeability, which can lead to the release of intracellular compounds. Depending on the fruit's structure, this may either increase the availability of antioxidants or cause their degradation.

Page1, line52-53

Line119-130: why are amino acids and sugar called metabolites? What is the reason for the relationships described in this paragraph? Is it related to the higher extraction efficiency or to the lower water content of the products?

Response: Amino acids and sugars are referred to as metabolites because they are essential components of cellular metabolism and play crucial roles in various biochemical processes within the body.

To show that previous studies had suggested that PEF could influence the metabolism of apricot.

It is related to higher extraction efficiency

Page3, line119-120

Material and methods
Line 148-159: why were these PEF parameters used? Were they determined on the basis of literature data or own experience?

Response: The induction of the stress response was achieved with intensities of 0.5-1.5 kV/cm and energy inputs of 0.5-5 kJ/kg. The improvement of mass transfer was targeted with in-tensities of 0.7-3 kV/cm and energy inputs of 1-20 kJ/kg(Toepfl, Mathys, Heinz, & Knorr, 2006). This information has been added to the Page4, line152-156 which reads.

These PEF parameters were chosen based on the desired effect of different PEF intensities. The induction of the stress response was achieved with intensities of 0.5-1.5 kV/cm and energy inputs of 0.5-5 kJ/kg. The improvement of mass transfer was targeted with intensities of 0.7-3 kV/cm and energy inputs of 1-20 kJ/kg(Toepfl et al., 2006).”

Line 162-165: How long did the freeze-drying process take? What was the vacuum and temperature of the sublimation process?

 Response: The freeze drying process took 72h with 0.05mbar and sublimation process around 0C with an additional secondary drying of 30C. This details has been added to Page 4 - line 172-175 which reads.

"The entire procedure lasted for 72 hours, with the vacuum set at 0.05 mbar. During the sublimation phase, the temperature was maintained at approximately 0°C, while in the secondary drying phase, the temperature was increased to 30°C."

Line 190-193: was the water content of the apricots determined on the basis of fruit subjected to sublimation or air drying?

Response: Yes after drying process finished. Page5, line193

Results and discussion
Line 304: instead of less should be higher.

Response: The text has been changed to higher at the  Page7, line 297-299 which reads.

“Specifically, the moisture content and water activity values of air-dried samples were found to be, on average, 28% and 30% higher than those of freeze-dried samples, respectively.”

Line 311-315: incorrectly described results for moisture content and aw for sublimation and air-dried samples.

Response: The reversed result has been changed back at the page7 line 306-310, which reads.

“For the samples subjected to air-drying, the moisture content and water activity values ranged between 27.231% to 27.47% and 0.544 to 0.571, respectively. For the samples subjected to freeze-drying, the moisture content and water activity values ranged between 19.42% to 19.853% and 0.384 to 0.399, respectively.”

Table 3 is very difficult to read. Please describe column 1 correctly. Please separate the results into 3 tables: 1. moisture and aw content; 2. texture characteristics; 3. antioxidant properties and polyphenol content.

Response: Table3 has been separated to table2,3&4 at page changed Page9-12, line318-335

Please correct the numbering of the subsections.

Response: Numbering of the subsections has been corrected.

Please describe this section in more detail: "However, for all the texture parameters, there were no significant differences between air-dried and freeze-dried control samples". Differences did occur. There should be no bracket when quoting: Acevedo, Briones, Buera, & Aguilera.

Response: The whole paragraph has been rewrote page13, line 337-386

Antioxidant properties: are the higher values after PEF treatment due to an increase in extraction efficiency or the formation of new compounds with higher antioxidant potential? Or rearrangements of existing compounds?

Response: The intensity and duration of the PEF treatment play a significant role. High-intensity treatments can cause greater cell disruption, potentially releasing more antioxidants or leading to oxidative stress, which may degrade antioxidants. The information has been added to the Page15, line 447-449 which reads.

“High-intensity treatments can cause greater cell disruption, potentially releasing more antioxidants or leading to oxidative stress, which may degrade antioxidants”

Metabolite profiling: figures 2 3 and are not very clear, they are unreadable. Please move the description of figure 2 below or above the figure.

Response: Added new figure Page 17-18, line 472-475

Table 4: very difficult to read. Please separate the results into 3 tables: 1. amino acid content; 2. fatty acid content; 3. organic acid content.

Response: Add in appendix Page28-31, line 842-859
Is it not possible to convert the amino acids to present them in other units in the table?

Response: New table make and added in appendix Page28-31, line 842-859

Please state from what you derive the relationships described in the paragraph: "In cluster 1 shown in Figure 3c, sugar content in air dried sample showed significantly p<0.05) higher sugar content than freeze dried sample".

Response: Its auto cluster by the metaboanalyst 4.0  Page 17-18, line 472-475

Conclusion
Please write, based on the results obtained, which treatment could be recommended (both PEF and drying method) for the most beneficial preservation of apricots.

Response: The conclusion part has been rewritten Page24, line 647-663

Reviewer 3 Report

Comments and Suggestions for Authors

The research presented has a limited interest and describes quite basic determinations of quality attributes of apricots pre-treated with pulsed electric field prior to drying operation.

Although the description of the methodologies is clear and the presentation of the results is relatively adequate, I believe the interest for readers might be limited.

However, the article was prepared with apparently accurate information and the discussion of results was also relatively pertinent. Still I do not understand the high relevance of amino-acids or fatty acids contents in fruits, where the relative amounts of proteins and fat are residual, compared to major components in fruits, such as sugars, for example.

Still, with drying it is lost a very high amount of water and the relative concentrations of other minority components may become a little more relevant.

Nevertheless, there is not much rationale in the experimental design and that diminished the interest of the article.

The use of English needs some improvement, definitely.

Comments on the Quality of English Language

The use of English need improvement.

Author Response

Reviewer 3

The research presented has a limited interest and describes quite basic determinations of quality attributes of apricots pre-treated with pulsed electric field prior to drying operation.

Although the description of the methodologies is clear and the presentation of the results is relatively adequate, I believe the interest for readers might be limited.

However, the article was prepared with apparently accurate information and the discussion of results was also relatively pertinent. Still I do not understand the high relevance of amino-acids or fatty acids contents in fruits, where the relative amounts of proteins and fat are residual, compared to major components in fruits, such as sugars, for example.

Response: Amino acids and fatty acids play significant roles in fruits, even if their concentrations are relatively low compared to major components like sugars. The high relevance of these metabolites lies in their influence on flavor, aroma, and other sensory characteristics. Here's why amino acids and fatty acids are important in fruits.

Amino acids and fatty acids are crucial precursors for various flavor and aroma compounds, contributing to the overall sensory profile of fruits. Through complex biochemical pathways, these metabolites can give rise to key flavor components.

Amino Acids: They can undergo enzymatic transformations to produce various aromatic compounds. For example, amino acids can be converted into alcohols, aldehydes, and esters .

Fatty Acids: Though present in smaller quantities, fatty acids can be oxidized to form volatile compounds that significantly impact flavor and aroma. The breakdown of fatty acids can lead to aldehydes, ketones, and other volatile compounds, adding complexity to the sensory perception of fruits.

Still, with drying it is lost a very high amount of water and the relative concentrations of other minority components may become a little more relevant.

Nevertheless, there is not much rationale in the experimental design and that diminished the interest of the article.

Round 2

Reviewer 1 Report

Comments and Suggestions for Authors

Most of my main concerns have been addressed in this review. However, still some corrections need to be made:

Line 152. Please report the pulse width

Line 162. report the timing between PEF and freezing

Lines 352-354. The text highlighted in yellow does not discuss the result. It does not discuss the result showing that air-dried samples have higher values of the texture parameters than freeze-dried samples. Of course PEF does not "improve the permeabilization" of cell membranes. PEF "increases the permeability" of the cell membrane.

Line 465. PEF permeabilizes cell membranes, not the cell wall.

Lines 481-498. Seems to be a mix-up between the text corresponding to the Figure legend and the text corresponding to the body of the article. Please check.

Author Response

Most of my main concerns have been addressed in this review. However, still some corrections need to be made:

Line 152. Please report the pulse width

Response: Pulse width was 20us. This information has been added to page 4 line 152, write as

“with a frequency of 50 Hz, and electric pulses with was 20 µs for 30 s.”

Line 162. report the timing between PEF and freezing

Response: Directly after PEF the samples were transferred to a freezer. The information on Page 4 line 162-164 has been rewritten as follows:

“For freeze drying, the apricot samples were frozen directly after PEF treatment using a Multifresh blast chiller (Irinox, USA) for 12 hours at -20 °C”.

Lines 352-354. The text highlighted in yellow does not discuss the result. It does not discuss the result showing that air-dried samples have higher values of the texture parameters than freeze-dried samples. Of course PEF does not "improve the permeabilization" of cell membranes. PEF "increases the permeability" of the cell membrane.

Response: The new discussion had been added to the page13 line 354-355, and written as:

In air-dried apricots, the enhanced water removal due to increased membrane permea-bility that could lead to a denser, harder texture as cells collapse more fully. In freeze-dried apricots, despite increased membrane permeability, the quick transition from frozen to dry minimizes the time for significant cellular collapse, hence maintaining a softer texture.

Line 465. PEF permeabilizes cell membranes, not the cell wall.

Response: Cell wall has been changed to cell membranes at page 16 line465-467, write as

“PEF treatment can help improve the extraction of TPC from date palm fruits due to the permeabilization of cell membrane, which can enhance mass transfer”

Lines 481-498. Seems to be a mix-up between the text corresponding to the Figure legend and the text corresponding to the body of the article. Please check.

Response: The Figure legend and text had been separated. at page 19 line484-485

Reviewer 2 Report

Comments and Suggestions for Authors

The manuscript has been revised in accordance with the reviewer's comments. I have two minor comments:
1. please add the results in quantitative terms to the abstract
2. line 41: it should be: Apricots are rich in vitamin A precursors.

Author Response

The manuscript has been revised in accordance with the reviewer's comments. I have two minor comments:
1. please add the results in quantitative terms to the abstract

Response :The quantitative result has been added. on page1 line 17-26.

Its not possible to include all the quantitative results for metabolites in the abstract.

2. line 41: it should be: Apricots are rich in vitamin A precursors.

Response: The text has been changed on page1 line 42 . This has been changed to:

Apricots are rich in vitamin A precursors (Ali et al., 2011).